# Spectral Assignment in the [3 + 2] Cycloadditions of Methyl (2*E*)-3-(Acridin-4-yl)-prop-2-enoate and 4-[(*E*)-2-Phenylethenyl]acridin with Unstable Nitrile N-Oxides

**DOI:** 10.3390/molecules29122756

**Published:** 2024-06-09

**Authors:** Lucia Ungvarská Maľučká, Mária Vilková

**Affiliations:** 1Institute of Chemistry, Faculty of Science, Pavol Jozef Šafárik University, Moyzesova 11, 040 01 Košice, Slovakia; lucia.ungvarskamalucka@uvlf.sk; 2Department of Chemistry, Biochemistry and Biophysics, The University of Veterinary Medicine and Pharmacy in Košice, Komenského 73, 041 81 Košice, Slovakia

**Keywords:** [3 + 2] cycloadditions, regioselectivity explanation, structure elucidation, geometric isomers resolution, relative configuration determination, 1D and 2D NMR spectroscopy

## Abstract

The investigation of cycloaddition reactions involving acridine-based dipolarophiles revealed distinct regioselectivity patterns influenced mainly by the electronic factor. Specifically, the reactions of methyl-(2*E*)-3-(acridin-4-yl)-prop-2-enoate and 4-[(1*E*)-2-phenylethenyl]acridine with unstable benzonitrile N-oxides were studied. For methyl-(2*E*)-3-(acridin-4-yl)-prop-2-enoate, the formation of two regioisomers favoured the 5-(acridin-4-yl)-4,5-dihydro-1,2-oxazole-4-carboxylates, with remarkable exclusivity in the case of 4-methoxybenzonitrile oxide. Conversely, 4-[(1*E*)-2-phenylethenyl]acridine displayed reversed regioselectivity, favouring products 4-[3-(substituted phenyl)-5-phenyl-4,5-dihydro-1,2-oxazol-4-yl]acridine. Subsequent hydrolysis of isolated methyl 5-(acridin-4-yl)-3-phenyl-4,5-dihydro-1,2-oxazole-4-carboxylates resulted in the production of carboxylic acids, with nearly complete conversion. During NMR measurements of carboxylic acids in CDCl_3_, decarboxylation was observed, indicating the formation of a new prochiral carbon centre C-4, further confirmed by a noticeable colour change. Overall, this investigation provides valuable insights into regioselectivity in cycloaddition reactions and subsequent transformations, suggesting potential applications across diverse scientific domains.

## 1. Introduction

Organic chemistry continually advances as researchers explore novel synthetic methodologies and elucidate reaction mechanisms with ever-increasing precision. Among the myriad of chemical transformations, [3 + 2] cycloadditions [1,2,3,4] stand as a cornerstone in constructing complex molecular structures. These reactions have garnered substantial attention due to their ability to prepare 4,5-dihydroisoxazoles. They can be reduced to several synthetically important compounds such as β-hydroxy ketones, β-hydroxy esters, α,β-unsaturated carbonyl compounds or iminoketones [5,6]. Cycloadditions with monosubstituted alkenes proceed rapidly and regioselectively [7,8,9,10]. On the other hand, the reaction of nitrile oxides with non-activated 1,2-disubstituted alkenes tends to be slower and usually affords mixtures of regio- and stereoisomers. However, in the case of activated alkenes, such as nitroalkenes, fully regioselective [3 + 2] cycloadditions are possible [11,12,13,14,15]. As a consequence, regioselectivity has been investigated extensively and a lot of attempts to control regioselectivity using chiral alkenes were described [16,17]. Computational studies of [3 + 2] cycloadditions have been carried out to rationalise the reactivity and regioselectivity of these reactions [18,19,20,21,22]. However, these results often contradict the experimentally observed findings [23].

Our previous investigation [13,14] unveiled the fascinating chemistry of acridine-alkene when paired with relatively stable nitrile N-oxides. The outcomes were not only synthetically valuable but also provided insights into the mechanistic intricacies of such transformations. Building upon this foundation, the present study represents a natural extension, delving deeper into the reactivity and spectral nuances encountered when these acridine dipolarophiles meet their less-stable counterparts—unstable nitrile N-oxides.

Unstable nitrile N-oxides have remained relatively underexplored in [3 + 2] cycloaddition chemistry, primarily due to their rapid dimerization, which brings with it challenges in their creation and manipulation. Because of the rapid dimerization, unstable nitrile N-oxides are usually synthesised in situ from hydroximoyl halides [24,25], aldoximes or from primary nitroalkanes [26,27], which often limits their use in organic synthesis. The inherent reactivity of unstable nitrile oxides poses intriguing questions: How does their fleeting existence influence the outcome of the cycloaddition? What insights can we gain from the elucidation of their reaction pathways?

In this context, nuclear magnetic resonance (NMR) spectroscopy emerges as an invaluable tool. It allows us to scrutinise the subtle spectral changes. By employing these advanced analytical methods, we aim to shed light on the unique challenges and opportunities presented by this intriguing class of compounds.

This study represents not only a continuation of our prior work but also a significant stride toward comprehending the broader landscape of [3 + 2] cycloadditions with nitrile N-oxides. The insights gained herein promise to enrich our understanding of the reactivity patterns of acridine-based systems and pave the way for the development of innovative synthetic strategies in organic chemistry.

## 2. Result and Discussion

### 2.1. [3 + 2] Cycloaddition Reactions of Methyl (2E)-3-(Acridin-4-yl)-acrylate (***1***) and 4-[(E)-2-Phenylethenyl]acridin (***2***) with Nitrile N-Oxides ***4a**–**e***

Our investigation focused on the cycloaddition reactions of acridine-alkenes, namely methyl (2*E*)-3-(acridin-4-yl)-prop-2-enoate (**1**) and 4-[(1*E*)-2-phenylethenyl]acridine (**2**) [13], with unstable nitrile N-oxides **4a**–**e**. Both alkene substrates **1** and **2** possess electron-accepting substituents. Identical reaction conditions were maintained across all ten reactions. The reactions involved dissolving acridine-alkene **1** or **2** in ethanol and subsequently adding a six-fold excess of the precursor of the three-atom-component (TAC)-*N*-hydroxybenzenecarbonimidoyl chloride **3a**–**e**. As the major limitation of the chemistry of isoxazolines is the propensity of nitrile N-oxides to undergo rapid dimerization to furoxan *N*-oxides [28,29], we circumvent this problem by generating the nitrile N-oxide species in situ, and an ethanolic solution of triethylamine was added dropwise over 8 days to generate the corresponding nitrile N-oxide **4a**–**e** (Figure 1). Benzonitrile N-oxide **4** partially reacted with dipolarophiles, the remainder dimerized into furoxane. Experiments demonstrated that the use of less than a six-fold excess of oxime **3** led to exceedingly slow conversion to cycloadducts. To maintain a sufficiently low concentration of nitrile oxide **4a**–**e** during the slow cycloaddition reaction, triethylamine had to be added very slowly. Consequently, the concentration of the generated unstable nitrile N-oxide **4a**–**e** remained only slightly more than that of the alkene **1** or **2** throughout the experiment.

Remarkably, our approach achieved nearly complete conversion of alkenes **1** or **2** to cycloadducts without the need for specific optimization of nitrile N-oxide **4a**–**e** formation for cycloaddition rates versus dimerization.

The progression of the cycloaddition was monitored using ^1^H NMR spectra. These spectra revealed the complete absence of alkenes **1** and **2**, confirming their near 100% conversion. Nonetheless, two pairs of doublets in the 4.00–7.00 ppm range, corresponding to the methine protons H-4 and H-5 of new regioisomers **5**/**6** and **7**/**8**, were observed. 

High regioselectivity of [3 + 2] cycloaddition reactions of **1** with nitrile oxides was observed. The ratios of regioisomers **5** and **6**, formed from alkene **1**, favoured the 5-(acridin-4-yl)-4,5-dihydro-1,2-oxazole-4-carboxylates **6a**,**b**,**d**,**e**. Notably, the ^1^H NMR spectra of the crude reaction mixture of alkene **1** with 4-methoxybenzonitrile oxide (**4a**) showed only one pair of isoxazoline CH doublets, indicating exclusive formation of the regioisomeric derivative **6a**, with no evidence of product **5a** (Table 1). 

In contrast, no regioselectivity was reversed in the reactions of alkene **2** with nitrile oxides **4a**–**e**. The reactions slightly favour the formation of products **7c**–**e**. Interestingly, a major **8a** isoxazoline was observed in the reaction of **2** with **4a**, indicating a distinct regioselectivity pattern.

It is well known that the regioselectivity of [3 + 2] cycloadditions depends on both steric and electronic effects. We propose that the regioselectivity of described reactions is highly governed by the electronic factors of the alkene which can be found from the magnitudes of their ^13^C chemical shifts. In the case of (2*E*)-3-(acridin-4-yl)-prop-2-enoate (**1**), the presence of a mildly polar acridine ring and a strongly electron-accepting methoxycarbonyl group results in a highly polarized C3=C2 double bond with chemical shifts 141.8 ppm for C-3 (closer to acridine) and 120.3 ppm for C-2 (distant from acridine). This polarization facilitates an attack by the nitrile oxide oxygen, thereby favouring the formation of product **6**. Conversely, in 4-[(1*E*)-2-phenyletenyl]acridine (**2**), the non-polar C1=C2 double bond with chemical shifts 130.5 ppm for C-1 (distant from acridine) and 125.2 ppm for C-2 (closer to acridine), owing to substituents with equivalent electron effects, leads to nearly equal preferences for either regioisomer (Table 1) [6,14]. 

The separation of cycloadducts from the reaction mixture proved to be labourious and time-consuming. Small quantities of sufficiently pure isoxazolines **5b**, **6a**,**b**,**d**,**e**, **7b**, **8a**–**c** for NMR analysis through multiple-column chromatography separations were obtained. However, it was unable to isolate adequate amounts of isoxazolines **5a**,**d**,**e**, and **7a**,**d**,**e**. Consequently, high-quality NMR spectra for these compounds could not be obtained, and their full characterisation remained elusive. However, their existence was confirmed by observing their presence in the ^1^H NMR spectra of the reaction mixture. Regioisomeric isoxazolines **7d**/**8d** and **7e**/**8e** could not be separated, and their structures were determined from the NMR spectra of these mixtures.

The complete structural characterisation of isolated compounds was achieved by the combined use of 1D and 2D NMR techniques (NMR spectra are included in Appendix A). The procedures used to assign NMR data to compounds **7d** and **8d** are explained in detail here. The ^1^H NMR spectrum (Figure 1) suggested the presence of two acridin-4-yl rings, two 1,3-disubstituted phenyl units, three types of protons with doublet-shaped signals, and one proton with a broad singlet-shaped signal. 

To distinguish between regioisomers **7d** and **8d**, ^13^C chemical shifts, as well as ^1^H,^13^C-HMBC correlations were analysed. ^13^C NMR signals of carbon atoms C-5 and C-4 of one regioisomeric isoxazoline were shown at 91.8 ppm and 55.7 ppm. This pair of carbons C-5 and C-4 provided HSQC correlations to protons H-5 at 5.79 ppm and H-4 at 6.58 ppm. Of note is a higher frequency of proton H-4 due to the magnetic anisotropy of neighbouring acridin-4-yl and phenyl moieties [15]. In both regioisomers *trans* relationship of protons H-4 and H-5 reflects the *E* configuration of the dipolarophiles **1** and **2** preserved in the cycloaddition reaction. ^13^C NMR signals of carbon atoms C-5 and C-4 of second regioisomeric isoxazoline were shown at 89.9 ppm and 62.5 ppm. This pair of carbons C-5 and C-4 provided HSQC correlations to protons H-5 at 6.78 ppm and H-4 at 4.92 ppm. To distinguish the regioisomers **7d** and **8d**, the HMBC correlations were crucial, namely, in regioisomer **7d** HMBC, correlations between proton H-5 (5.79 ppm) and carbon atoms C-4′ (136.4 ppm) and C-2″,6″ (125.7 ppm), and in regioisomer **8d** HMBC, correlations between proton H-4 (4.92 ppm) and carbon atoms C-4′ (137.5 ppm) and C-2″,6″ (128.3 ppm) (Figure 2).

Analysis of 2D COSY, 2D NOESY and 1D TOCSY spectra as well as 2D HSQCTOXY allowed resolution of the protons of six spin systems, two acridin-4-yl rings, and two 1,3-disubstituted phenyl units. The starting point for the assigning signals for both acridin-4-yl moieties were NOESY correlations between one proton singlets of H-9′ (**7e**: 8.84 ppm; **8e**: 8.81 ppm) and protons H-1′ (**7d**: 7.99 ppm; **8d**: 7.97 ppm) and H-8′ (**7d**: 8.06 ppm; **8d**: 8.03 ppm) (Figure 3). The signals of the protons and carbons 1′–3′ and 5′–8′ for both acridin-4-yl systems as well as the signals of the protons and carbons 2″,4″–6″ for both phenyl rings were assigned using homonuclear ^1^H,^1^H-COSY correlations, heteronuclear ^1^H,^13^C-HSQCTOXY, ^1^H,^13^C-HSQC correlations, and ^1^H,^13^C-HMBC correlations. The ^1^H,^13^C-HMBC spectra allowed for assigning acridine quaternary carbons C-4′, C-8′a, C-9′a, C-4′a, and C-10′a, phenyl quaternary carbons C-1″, C-4″ and C-1‴, C-3‴, and quaternary carbon of isoxazoline fragment C-3 (Figure 4).

The absolute configuration of C-4 and C-5 stereogenic centres in all regioisomers was not determined. 

### 2.2. Basic Hydrolysis of ***6a**,**b**,**d**,**e*** and Formation of Carboxylic Acids ***9a**,**b**,**d**,**e*** and Isoxazole-5-ones Z-***10e*** and E-***10e***

The subsequent step involved the hydrolysis of isolated esters **6a**,**b**,**d**,**e** to produce carboxylic acids **9a**,**b**,**d**,**e**. These reactions were carried out in ethanol with a 10-fold excess of the base over 4 h, resulting in nearly 100% conversion of starting substances **6a**,**b**,**d**,**e**. Hydrolysis of derivative **6e** yielded three products: acid **9e** (81%) and two isoxazole-5-one stereoisomers, *Z*-**10e** (14%) and *E*-**10e** (5%) (Figure 2) [13]. 

The ^1^H chemical shift, splitting patterns, and intensities of proton signals for derivatives **9a**,**b**,**d**,**e** were consistent with those of the starting esters **6a**,**b**,**d**,**e**. The only noticeable difference in all these substances was the absence of the ^1^H NMR singlet signal of the methyl ester group around 3.90 ppm. In addition, the ^13^C NMR spectra of all products **9a**,**b**,**d**,**e** exhibited a slight shift (approximately 0.9–2.1 ppm) of the C=O group signal to lower ppm values compared to the starting esters **6a**,**b**,**d**,**e**. 

The separation and purification of isoxazole-5-ones *Z*-**10e**, and *E*-**10e** proved challenging, yielding only a mixture of isoxazole-5-one *Z*-**10e** (87%) along with the *E*-**10e** derivative (13%) after repeated crystallization. While proton and carbon signals of the major *Z*-**10e** were successfully assigned based on NMR experiments, the minor derivative *E*-**10e** proton and carbon signals could not be assigned. The preliminary analysis of the ^1^H NMR spectrum of the mixture of stereoisomers *Z*-**10e** and *E*-**10e** in CDCl_3_ revealed signals with no overlap, facilitating the direct measurement of chemical shifts and *J* values and the correct determination of their multiplicities. The ^1^H and ^13^C NMR chemical shifts measured in CDCl_3_ are in reasonable agreement with those measured previously [13].

The acridine proton–proton connectivity was traced starting from NOESY correlations between proton H-9′ and protons H-1′/H-8′. The standard gCOSY experiment revealed H1′–H3′ and H5′–H8′ connectivities. The chemical shifts of protons attached to acridine carbon atoms were assigned through a straightforward application of the gHSQC experiment. The nonprotonated carbons C-4′, C-4′a/C-10′a, and C-8′a/C-9′a of acridin-4-yl moiety were assigned using their HMBC connectivities with the protons three-bonds distant. Additionally, the assignments of unprotonated carbons C-3, C-4, and C-5 of isoxazolone moiety were unequivocally accomplished through their observed HMBC connectivities with H-6 and H-2″,6″ protons (Figure 5 and Figure 6) [6]. 

For isoxazole-5-one *Z*-**10e**, the magnetic anisotropy effect on the acridine proton H-3’s chemical shift was evident. The high chemical shift (9.73 ppm) of the proton doublet H-3′ can be attributed to the magnetic anisotropic effect of the spatially close C=O group. In addition, a synergic effect of two electron-acceptor groups, C=O and C=N, elicited a strong deshielding of proton H-6 to 9.85 ppm.

It appears that the possibility of isoxazolone formation depends on the electron-withdrawing character of the phenyl substituent. The electron-withdrawing nitro group favoured the formation of isoxazole-5-one **10e**, while the presence of unsubstituted phenyl, the electron-donor methoxy group or the nitro group in position 3 on the phenyl ring inhibited the formation of isoxazole-5-one. 

### 2.3. Carboxylic Acids ***9a**,**b**,**d**,**e*** Decarboxylation and Formation of 4-(3-Phenyl-4,5-dihydro-1,2-oxazol-5-yl)acridines ***11a**,**b**,**d**,**e***

During NMR measurements of carboxylic acids **9a**,**b**,**d**,**e** in CDCl_3_, it was observed that decarboxylation occurred, resulting in NMR spectra featuring two distinct sets of signals (Figure 7). The most prominent indication of decarboxylation was the appearance of three new signals corresponding to the protons H-4a, H-4b and H-5. With the formation of the new prochiral carbon centre C-4, the two signals for protons H-4a and H-4b became non-equivalent. The occurrence of decarboxylation was further evidenced by a noticeable colour change from light yellow to dark green [30].

The structures of the non-purified decarboxylated products **11a**,**b**,**d**,**e** were elucidated using 1D and 2D (TOCSY, H2BC, HMBC) NMR spectra and compared with those of carboxylic acids **9a**,**b**,**d**,**e**. The ^1^H chemical shift, splitting patterns, and intensities of proton signals for derivatives **11a**,**b**,**d**,**e** were consistent with those of the starting acids **9a**,**b**,**d**,**e**. The only noticeable difference in all these substances was the presence of three signals corresponding to protons H-4a, H-4b and H-5, with chemical shifts of 4.20 ppm, 3.40 ppm, and 6.90 ppm, respectively. The relative stereochemistry of the prochiral carbon centre C-4 was determined through 2D NOESY spectra, where a NOESY cross peak between protons H-5 (6.9 ppm) and H-4a (4.2 ppm) was observed as well as through homonuclear coupling constants (Figure 8). As was written by Thomas, it is clear that as well as the dependence on dihedral angle, vicinal coupling constants depend on the electronegativity and orientation of substituents on the H–C–C–H fragment with both α and β effects, the H–C–C bond angles, overlap of orbitals from adjacent nuclei, and possibly on lone pairs and hyperconjugative effects. The lone pairs on nitrogen and oxygen have specific effects on both chemical shifts and coupling constants for protons on adjacent carbons [25]. The coupling constant ^3^*J* for the protons H-5 and H-4a, situated on the opposite side of the five-membered isoxazoline ring, falls within the 11.2–11.5 Hz range. However, the coupling constant ^3^*J* between protons H-5 and H-4b on the same side is in the 7.5–7.9 Hz range. 

## 3. Experimental Section

### 3.1. General

All reagents (Merck, Darmstadt, Germany) were used as supplied without prior purification. The progression of the reaction was monitored by analytical thin-layer chromatography using TLC sheets ALUGRAM-SIL G/UV254 (Macherey Nagel, Düren, Germany). Purification by flash chromatography was performed using silica gel (60 Å, 230–400 mesh, Merck, Darmstadt, Germany) with the indicated eluent. 

### 3.2. Melting Point Determination

The melting points of the synthesised derivatives were determined using a Stuart^TM^ melting point apparatus SMP10 (Bibby Scientific Ltd., Staffordshire, UK). 

### 3.3. NMR Spectroscopy

NMR spectra were acquired using a Varian VNMRS spectrometer (Palo Alto, CA, USA) operating at 599.87 MHz for ^1^H, 150.84 MHz for ^13^C, and Varian Mercury spectrometer (Palo Alto, CA, USA) operating at 400.13 MHz for ^1^H and 100.62 MHz for ^13^C. These experiments were conducted at a temperature of 299.15 K, and a 5 mm inverse-detection H-X probe with a z-gradient coil was used. Pulse programs from the Varian sequence library were employed. Chemical shifts (*δ* in ppm) were referenced to internal solvent standard CDCl_3_ 77.0 ppm for ^13^C, while a partially deuterated signal of CHD_2_Cl 7.26 ppm was used for ^1^H referencing. MestReNova v. 15.0.1 (Mestrelab Research, Santiago de Compostela, Spain) was utilized for NMR spectra processing and analysis.

### 3.4. IR Spectroscopy

The infrared spectra of prepared compounds were recorded with Avatar FT−IR 6700 (Fourier transform infrared spectroscopy) spectrometer in the range from 400 to 4000 cm^−1^ with 64 repetitions for a single spectrum using the ATR (attenuated total reflectance) technique. All obtained data were analysed using Omnic 8.2.0.387 (2010) software, and the structure of all new compounds was confirmed by analysis of FT-IR spectrum by functional group identification. 

### 3.5. Elemental Analysis

Elemental analysis of C, H, and N was performed using a CHNOS Elemental Analyzer vario MICRO from Elementar Analysensysteme GmbH (Langenselbold, Germany).

### 3.6. General Procedure for Preparation of Methyl 4-(Acridin-4-yl)-3-phenyl-4,5-dihydro-1,2-oxazole-5-carboxylates ***5a**,**b**,**d**,**e*** and Methyl 5-(Acridin-4-yl)-3-phenyl-4,5-dihydro-1,2-oxazole-4-carboxylates ***6a**–**e***

The corresponding *N*-hydroxybenzenecarbonimidoyl chloride (**3a**: 423 mg; **3b**: 354 mg; **3c**: 533 mg; **3d**: 457 mg; **3e**: 457 mg, 2.28 mmol) was added to an ethanolic solution (5 mL) of methyl (2*E*)-3-(acridin-4-yl)-prop-2-enoate (**1**, 100 mg, 0.37 mmol), and the reaction mixture was heated to 40 °C. Triethylamine (230 mg, 0.317 mL, 2.28 mmol) was dissolved in ethanol (5 mL) and added to the reaction mixture over eight days. The reaction’s progress was tracked using ^1^H NMR spectra. Further purification by column chromatography (SiO_2_, *n*-Hex/EtOAc, 5:1) yielded compounds **5b** and **6a**,**b**,**d**,**e**.

Methyl-4-(acridin-4-yl)-3-phenyl-4,5-dihydro-1,2-oxazole-5-carboxylate (**5b**). Yield: 7.0 mg (5.0%). Mp. 114–116 °C. Yellow needles. For C_24_H_18_N_2_O_3_ (382.13) found: C 74.78, H 5.30, N 7.21%; calc.: C 75.38, H 4.74, N 7.33, O 12.55%. *R_f_* (*n*-Hex/EtOAc, 5:1) 0.13. FT-IR: ν_max_ 3104, 3069, 1782, 1647, 1218, 1167, 1025, 753 cm^−1^. ^1^H NMR (600 MHz CDCl_3_): δ 8.80 (1H, s, H-9′), 8.29 (1H, dd, *J* = 8.4, 1.2 Hz, H-5′), 8.04 (1H, d, *J* = 8.4 Hz, H-8′), 7.95 (1H, dd, *J* = 8.4, 1.2 Hz, H-1′), 7.84 (1H, ddd, *J* = 8.4, 6.6, 1.8 Hz, H-6′), 7.73 (2H, dd, *J* = 6.6, 1.2 Hz, H-2″,6″), 7.60 (1H, ddd, *J* = 8.4, 6.6, 1.2 Hz, H-7′), 7.54 (1H, dd, *J* = 6.6, 1.2 Hz, H-3′), 7.42 (1H, dd, *J* = 8.4, 7.2 Hz, H-2′), 7.27 (1H, t, *J* = 8.4 Hz, H-4″), 7.21 (2H, t, *J* = 7.8 Hz, H-3″,5″), 6.85 (1H, d, *J* = 4.2 Hz, H-4), 5.05 (1H, d, *J* = 4.2 Hz, H-5), 3.91 (3H, s, H-7) ppm. ^13^C NMR (150.1 MHz, CDCl_3_): δ 170.5 (C-6), 159.4 (C-3), 148.7 (C-10′a), 146.0 (C-4′a), 136.3 (C-9′), 136.2 (C-4′), 130.5 (C-6′), 130.1 (C-4″), 130.1 (C-5′), 128.6 (C-3″,5″), 128.5 (C-1′), 128.3 (C-1″), 128.0 (C-8′), 127.7 (C-2″,6″), 126.8 (C-3′,C-8′a), 126.7 (C-9′a), 126.2 (C-7′), 125.5 (C-2′), 86.7 (C-5), 52.7 (C-7), 52.0 (C-4) ppm.

Methyl-5-(acridin-4-yl)-3-(4-methoxyphenyl)-4,5-dihydro-1,2-oxazole-4-carboxylate (**6a**). Yield: 133.1 mg (85.0%). M.p. 152–154 °C. Yellow powder. For C_25_H_20_N_2_O_4_ (412.45) found: C 72.75, H 4.73, N 6.88%; calc.: C 72.80, H 4.89, N 6.79, O 15.52%. *R_f_* (5:1 *v*/*v n*-Hex/EtOAc) 0.16. FT-IR: ν_max_ 3102, 3063, 2944, 2903, 1768, 1647, 1621, 1482, 1338, 1250, 1172, 1065, 1028 cm^−1^. ^1^H NMR (400 MHz CDCl_3_): δ 8.78 (1H, s, H-9′), 8.09 (1H, d, *J* = 8.8 Hz, H-5′), 8.01 (1H, d, *J* = 8.8 Hz, H-8′), 7.98 (1H, d, *J* = 6.8 Hz, H-3′), 7.96 (1H, d, *J* = 8.0 Hz, H-1′), 7.78 (1H, ddd, *J* = 8.4, 6.4, 1.2 Hz, H-6′), 7.71 (2H, d, *J* = 8.8 Hz, H-2″,6″), 7.56 (1H, ddd, *J* = 8.8, 6.4, 1.2 Hz, H-7′), 7.53 (1H, dd, *J* = 8.0, 6.8 Hz, H-2′), 7.12 (1H, d, *J* = 6.0 Hz, H-5), 6.85 (2H, d, *J* = 8.8 Hz, H-3″,5″), 4.59 (1H, d, *J* = 6.0 Hz, H-4), 3.95 (3H, s, H-7), 3.80 (3H, s, OCH_3_) ppm. ^13^C NMR (100.6 MHz, CDCl_3_): δ 170.6 (C-6), 161.2 (C-4″), 153.8 (C-3), 148.1 (C-10′a), 145.9 (C-4′a), 137.4 (C-4′), 136.0 (C-9′), 130.3 (C-6′), 129.7 (C-5′), 128.6 (C-2″,6″), 128.1 (C-8′), 128.2 (C-1′), 126.6 (C-8′a), 126.66 (C-9′a), 126.0 (C-7′), 125.9 (C-3′), 125.4 (C-2′), 121.1 (C-1″), 114.1 (C-3″,5″), 84.8 (C-5), 62.6 (C-4), 55.3 (OCH_3_), 52.9 (C-7) ppm.

Methyl-5-(acridin-4-yl)-3-phenyl-4,5-dihydro-1,2-oxazole-4-carboxylate (**6b**). Yield: 116.1 mg (80.0%). Mp. 177–179 °C. Yellow powder. For C_24_H_18_N_2_O_3_ (382.13) found: C 74.80, H 5.22, N 7.30%; calc.: C 75.38, H 4.74, N 7.33, O 12.55%; *R_f_* (*n*-Hex/EtOAc, 5:1) 0.29. FT-IR: ν_max_ 3096, 2910, 1770, 1646, 1604, 1536, 1328, 1217, 1045 cm^−1^. ^1^H NMR (400 MHz CDCl_3_): δ 8.78 (1H, s, H-9′), 8.08 (1H, d, *J* = 8.8 Hz, H-5′), 8.01 (1H, dd, *J* = 8.5, 0.7 Hz, H-8′), 7.97 (1H, m, H-1′), 7.97 (1H, m, H-3′), 7.78 (1H, m, H-6′), 7.78 (2H, m, H-2″,6″), 7.56 (1H, ddd, *J* = 8.5, 6.6, 1.2 Hz, H-7′), 7.53 (1H, dd, *J* = 8.5, 6.9 Hz, H-2′), 7.36 (3H, m, H-3″,5″, H-4″), 7.15 (1H, d, *J* = 6.5 Hz, H-5), 4.62 (1H, d, *J* = 6.5 Hz, H-4), 3.95 (3H, s, H-7) ppm. ^13^C NMR (100.6 MHz, CDCl_3_): δ 170.5 (C-6), 154.2 (C-3), 148.1 (C-10′a), 145.8 (C-4′a), 137.2 (C-4′), 136.0 (C-9′), 130.4 (C-6′), 130.2 (C-4″), 129.7 (C-5′), 128.7 (C-3″,5″), 128.1 (C-1′), 128.2 (C-8′), 128.6 (C-1″), 127.0 (C-2″,6″), 126.6 (C-8′a,9′a), 126.0 (C-3′), 126.0 (C-7′), 125.4 (C-2′), 85.1 (C-5), 62.3 (C-4), 52.9 (C-7) ppm.

Methyl-5-(acridin-4-yl)-3-(3-nitrophenyl)-4,5-dihydro-1,2-oxazole-4-carboxylate (**6d**). Yield: 102.3 mg (63.0%). Mp. 113–115 °C. Yellow needles. For C_24_H_17_N_3_O_5_ (427.42) found: C 67.24, H 4.08, N 9.80%; calc.: C 67.44, H 4.01, N 9.83, O 18.27%. *R_f_* (5:1 *v*/*v n*-Hex/EtOAc) 0.16. FT-IR: ν_max_ 3098, 2910, 1780, 1663, 1337, 1231, 1172, 1165, 1071, 1058 cm^−1^. ^1^H NMR (400 MHz CDCl_3_): δ 8.80 (1H, s, H-9′), 8.62 (1H, d, *J* = 2.0 Hz, H-2″), 8.23 (1H, d, *J* = 8.4 Hz, H-4″), 8.17 (1H, d, *J* = 8.4 Hz, H-6″), 8.05 (1H, d, *J* = 8.4 Hz, H-5′), 8.03 (1H, d, *J* = 8.4 Hz, H-8′), 7.99 (1H, d, *J* = 8.4 Hz, H-8′), 7.96 (1H, dd, *J* = 6.8, 1.3 Hz, H-3′), 7.79 (1H, m, H-6′), 7.56 (3H, m, H-2′, H-7′, H-5″), 7.21 (1H, d, *J* = 6.8 Hz, H-5), 4.69 (1H, d, *J* = 6.8 Hz, H-4), 3.98 (3H, s, H-7) ppm. ^13^C NMR (100.6 MHz, CDCl_3_): δ 170.0 (C-6), 152.6 (C-3), 148.5 (C-3″), 148.1 (C-10′a), 145.7 (C-4′a), 136.6 (C-4′), 136.2 (C-9′), 132.6 (C-6″), 130.7 (C-1″), 130.5 (C-6′), 129.8 (C-5″),129.6 (C-5′), 128.6 (C-1′), 128.2 (C-8′), 126.6 (C-8′a), 126.6 (C-9′a), 126.2 (C-7′), 126.0 (C-3′), 125.3 (C-2′), 124.6 (C-4″), 121.9 (C-2″), 86.1 (C-5), 61.6 (C-4), 53.1 (C-7) ppm.

Methyl-5-(acridin-4-yl)-3-(4-nitrophenyl)-4,5-dihydro-1,2-oxazole-4-carboxylate (**6e**). Yield: 115.3 mg (71.0%). Mp. 165–167 °C. Yellow needles. For C_24_H_17_N_3_O_5_ (427.42) found: C 67.34, H 4.11, N 9.73%; calc.: C 67.44, H 4.01, N 9.83, O 18.27%. *R_f_* (CH_2_Cl_2_) 0.56. FT-IR: ν_max_ 3124, 3102, 2980, 1771, 1652, 1629, 1621, 1640, 1566, 1344, 1157, 1067, 1039, 842 cm^−1^. ^1^H NMR (400 MHz CDCl_3_): δ 8.79 (1H, s, H-9′), 8.21 (2H, d, *J* = 8.9, Hz H-3″,5″), 8.02 (3H, m, H-1′, H-5′, H-8′), 7.96 (3H, m, H-3′, H-2″,6″), 7.78 (1H, ddd, *J* = 8.4, 6.8, 1.2 Hz, H-6′), 7.57 (1H, ddd, *J* = 8.4, 6.8, 1.2 Hz, H-7′), 7.54 (1H, dd, *J* = 8.4, 6.8 Hz, H-2′), 7.18 (1H, d, *J* = 6.8 Hz, H-5), 4.68 (1H, d, *J* = 6.8 Hz, H-4), 3.97 (3H, s, H-7) ppm. ^13^C NMR (100.6 MHz, CDCl_3_): δ 169.9 (C-6), 152.9 (C-3), 148.5 (C-4″), 148.1 (C-10′a), 145.6 (C-4′a), 136.5 (C-4′), 136.2 (C-9′), 134.9 (C-1″), 130.5 (C-6′), 129.5 (C-5′), 128.6 (C-1′), 128.2 (C-8′), 127.8 (C-2″,6″), 126.6 (C-8′a, C-9′a), 126.2 (C-3′, C-7′), 125.2 (C-3′), 123.9 (C-3″,5″), 86.4 (C-5), 61.5 (C-4), 53.2 (C-7) ppm.

### 3.7. General Procedure for Preparation of 4-[3-(4-Nitrophenyl)-5-phenyl-4,5-dihydro-1,2-oxazol-4-yl]acridines ***7a**–**e*** and 4-(3-Phenyl-4-phenyl-4,5-dihydro-1,2-oxazol-5-yl)acridines ***8a**–**e***

A corresponding *N*-hydroxybenzenecarbonimidoyl chloride (**3a**: 395 m; **3b**: 331 mg; **3c**: 500 mg; **3d**: 427 mg; **3e**: 427 mg, 2.13 mmol) was added to a solution of methyl 4-[(1*E*)-2-phenylethenyl]acridine (**2**, 100 mg, 0.35 mmol) in ethanol (5 mL), and the reaction mixture was heated to 40 °C. Over eight days, a triethylamine solution (215 mg, 0.296 mL, 2.28 mmol) in ethanol (5 mL) was added. ^1^H NMR spectra of the reaction mixture were used to track the process. The solvent was evaporated under reduced pressure. Further purification by column chromatography (SiO_2_, *c*-Hex/EtOAc, 5:1) yielded compounds **7b** and **8b**, and (SiO_2_, *n*-Hex/EtOAc, 5:1) yielded compounds **8a**,**c** and the mixture of isomers **7d**/**8d** and **7e**/**8e**.

4-(3,5-Diphenyl-4,5-dihydro-1,2-oxazole-4-yl)acridine (**7b**). Yield: 55.5 mg (39.0%). Mp. 177–179 °C. Yellow crystals. For C_28_H_20_N_2_O (400.48) found: C 84.01, H 5.16, N 7.13%; calc.: C 83.98, H 5.03, N 6.99, O 4.00%. *R_f_* (5:1 *v*/*v n*-Hex/EtOAc) 0.35. FT-IR: ν_max_ 3091, 2895, 1640, 1355, 1034, 1010, 874, 754 cm^−1^. ^1^H NMR (400 MHz CDCl_3_): δ 8.81 (1H, s, H-9′), 8.29 (1H, dd, *J* = 8.8, 1.0 Hz, H-5′), 8.05 (1H, dd, *J* = 8.4, 1.0 Hz, H-8′), 7.95 (1H, dd, *J* = 8.6, 1.4 Hz, H-1′), 7.84 (1H, ddd, *J* = 8.4, 6.6, 1.4 Hz, H-6′), 7.78 (2H, d, *J* = 7.8 Hz, H-2″,6″), 7.69 (2H, dd, *J* = 8.4, 1.5 Hz, H-2‴,6‴), 7.60 (2H, m, H-3′, H-7′), 7.44 (3H, m, H-2′, H-3″,5″), 7.35 (1H, td, *J* = 8.5, 1.2 Hz, H-4″), 7.20 (3H, m, H-3‴,5‴, H-4‴), 6.52 (1H, d, *J* = 3.4 Hz, H-4), 5.64 (1H, d, *J* = 3.4 Hz, H-5) ppm. ^13^C NMR (100.6 MHz, CDCl_3_): δ 159.0 (C-3), 148.6 (C-10′a), 146.4 (C-4′a), 141.6 (C-1‴), 137.1 (C-4′), 136.3 (C-9′), 130.4 (C-6′), 129.8 (C-4‴, C-5′), 129.1 (C-1‴), 128.6 (C-3′), 128.4 (C-3‴,5‴), 128.5 (C-3″,5″), 128.1 (C-8′), 128.1 (C-1′), 127.7 (C-1″, C-4″), 127.5 (C-2‴,6‴), 126.9 (C-9′a), 126.7 (C-8′a), 126.1 (C-7′), 125.8 (C-2′), 125.8 (C-2″,6″), 90.9 (C-5), 56.4 (C-4) ppm.

4-[3-(3-Nitrophenyl)-5-phenyl-4,5-dihydro-1,2-oxazole-4-yl]acridine (**7d**). The compound **7e** was obtained as a mixture of **7d** and **8e**. *R_f_* (5:1 *v*/*v c*-Hex/EtOAc) 0.32. ^1^H NMR (600 MHz CDCl_3_): δ 8.84 (1H, s, H-9′), 8.64 (1H, t, *J* = 2.0 Hz, H-2‴), 8.31 (1H, dd, *J* = 8.8, 1.0 Hz, H-5′), 8.06 (1H, m, H-4‴, H-8′), 7.99 (2H, m, H-1′, H-6‴), 7.86 (1H, ddd, *J* = 8.8, 6.6, 1.4 Hz, H-6′), 7.70 (2H, d, *J* = 7.8 Hz, H-2″,6″), 7.61 (2H, m, H-3′, H-7′), 7.48 (1H, dd, *J* = 8.4, 7.0 Hz, H-2′), 7.46 (1H, t, *J* = 7.6 Hz, H-3″,5″), 7.38 (1H, td, *J* = 7.4, 1.8 Hz, H-4″), 7.35 (1H, t, *J* = 8.0 Hz, H-3‴,5‴), 6.58 (1H, br s, H-4), 5.78 (1H, d, *J* = 3.9 Hz, H-5) ppm. ^13^C NMR (150.1 MHz, CDCl_3_): δ 157.8 (C-3), 148.8 (C-10′a), 148.4 (C-3‴), 146.0 (C-4′a), 140.9 (C-1″), 136.6 (C-9′), 136.4 (C-4′), 131.0 (C-1‴), 132.9 (C-6″), 130.6 (C-6′), 129.9 (C-5′), 129.5 (C-5‴), 128.6 (C-3″,5″), 128.1 (C-4″), 128.6 (C-1′), 128.3 (C-3′), 128.0 (C-8′), 126.8 (C-9′a), 126.7 (C-8′a), 126.3 (C-7′), 125.6 (C-2′), 125.7 (C-2″,6″), 124.2 (C-4‴), 122.3 (C-2‴), 91.8 (C-5), 55.7 (C-4) ppm.

4-[3-(4-Nitrophenyl)-5-phenyl-4,5-dihydro-1,2-oxazole-4-yl]acridine (**7e**). The compound **7e** was obtained as a mixture of **7e** and **8e**. *R_f_* (5:1 *v*/*v c*-Hex/EtOAc) 0.32. ^1^H NMR (400 MHz CDCl_3_): δ 8.84 (1H, s, H-9′), 8.27 (2H, d, *J* = 9.0 Hz, H-3‴,5‴), 8.27 (1H, m, H-5′), 8.06 (1H, m, H-8′), 7.99 (1H, dd, *J* = 8.4, 1.0 Hz, H-8′), 7.85 (2H, d, *J* = 9.0 Hz, H-2‴,6‴), 7.85 (1H, m, H-6′), 7.71 (2H, m, H-2″,6″), 7.56 (1H, m, H-3′, H-7′), 7.46 (3H, m, H-2′, H-3″,5″), 7.38 (1H, m, H-4″), 6.55 (1H, br s, H-4), 5.76 (1H, d, *J* = 4.1 Hz, H-5) ppm. ^13^C NMR (100.6 MHz, CDCl_3_): δ 157.8 (C-3), 148.7 (C-10′a), 148.1 (C-4‴), 146.1 (C-4′a), 140.8 (C-1″), 136.4 (C-4′), 136.3 (C-9′), 135.4 (C-1‴), 130.7 (C-6′), 128.1 (C-4″), 129.7 (C-5′), 128.7 (C-3″,5″), 128.6 (C-1′), 128.2 (C-8′), 126.9 (C-9′a), 126.8 (C-8′a), 128.1 (C-2‴,6‴, C-3′), 126.3 (C-7′), 125.7 (C-2″,6″), 125.6 (C-2′), 123.5 (C-3‴,5‴), 92.0 (C-5), 55.7 (C-4) ppm.

4-[3-(4-Methoxyphenyl)-4-phenyl-4,5-dihydro-1,2-oxazole-5-yl]acridine (**8a**). Yield: 64.3 mg (42.0%). Mp. 225–227 °C. Yellow solid. For C_29_H_22_N_2_O_2_ (430.51) found: C 81.00, H 5.10, N 6.63%; calc.: C 80.91, H 5.15, N 6.51, O 7.43%. *R_f_* (5:1 *v*/*v c*-Hex/EtOAc) 0.32. FT-IR: ν_max_ 3085, 2930, 2915, 1638, 1608, 1564, 1480, 1353, 1264, 1205, 1056, 1019, 796, 754 cm^−1^. ^1^H NMR (400 MHz CDCl_3_): δ 8.77 (1H, s, H-9′), 8.12 (1H, d, *J* = 8.4 Hz, H-5′), 8.01 (1H, d, *J* = 8.4 Hz, H-8′), 7.96 (1H, dd, *J* = 6.8, 1.2 Hz, H-3′), 7.93 (1H, d, *J* = 8.5 Hz, H-1′), 7.79 (1H, ddd, *J* = 8.4, 6.8, 1.2 Hz, H-6′), 7.63 (2H, d, *J* = 8.2 Hz, H-2″,6″), 7.54 (1H, m, H-7′), 7.54 (3H, m, H-2‴,6‴, H-2′), 7.48 (2H, t, *J* = 7.6 Hz, H-3″,5″), 7.38 (1H, t, *J* = 7.6 Hz, H-4″), 6.72 (2H, d, *J* = 8.9 Hz, H-3‴,5‴), 6.68 (1H, d, *J* = 3.2 Hz, H-5), 4.84 (1H, d, *J* = 3.2 Hz, H-4), 3.70 (3H, s, OCH_3_) ppm. ^13^C NMR (100.6 MHz, CDCl_3_): δ 160.7 (C-4‴), 158.4 (C-3), 148.2 (C-10′a), 146.4 (C-4′a), 139.7 (C-1″), 138.2 (C-4′), 136.0 (C-9′), 130.2 (C-6′), 129.6 (C-5′), 128.8 (C-3″,5″, C-2‴,6‴), 128.3 (C-2″,6″), 128.2 (C-8′), 127.8 (C-1′), 127.5 (C-4″), 126.8 (C-8′a), 126.4 (C-9′a), 126.1 (C-3′), 125.8 (C-2′, C-7′), 121.6 (C-1‴), 114.0 (C-3‴,5‴), 88.6 (C-5), 63.1 (C-4), 55.2 (OCH_3_) ppm. 

4-(3,4-Diphenyl-4,5-dihydro-1,2-oxazole-5-yl)acridine (**8b**). Yield: 59.8 mg (42.0%). Mp. 210–212 °C. Yellow powder. For C_28_H_20_N_2_O (400.48) found: C 83.82, H 5.11, N 7.06%; calc.: C 83.98, H 5.03, N 6.99, O 4.00%. *R_f_* (5:1 *v*/*v* 5:1 *v*/*v c*-hex/EtOAc) 0.39. FT-IR: ν_max_ 3105, 2918, 1641, 1479, 1332, 1060, 1056, 877, 764 cm^−1^. ^1^H NMR (400 MHz CDCl_3_): δ 8.79 (1H, s, H-9′), 8.12 (1H, d, *J* = 8.8 Hz, H-5′), 8.03 (1H, dd, *J* = 8.4, 1.0 Hz, H-8′), 7.95 (2H, m, H-1′, H-3′), 7.79 (1H, ddd, *J* = 8.4, 6.8, 1.2 Hz, H-6′), 7.61 (4H, m, H-2″,6″, H-2‴,6‴), 7.53 (2H, m, H-2′, H-7′), 7.49 (2H, t, *J* = 7.8 Hz, H-3″,5″), 7.39 (1H, t, *J* = 7.6 Hz, H-4″), 7.23 (3H, m, H-3″,5″, H-4″), 6.72 (1H, d, *J* = 3.2 Hz, H-5), 4.87 (1H, d, *J* = 3.2 Hz, H-4) ppm. ^13^C NMR (100.6 MHz, CDCl_3_): δ 159.0 (C-3), 148.3 (C-10′a), 146.5 (C-4′a), 139.7 (C-1″), 138.2 (C-4′), 136.2 (C-9′), 130.3 (C-6′), 129.9 (C-4‴), 129.8 (C-5′), 129.3 (C-1‴), 129.1 (C-3″,5″), 128.7 (C-3‴,5‴), 128.5 (C-2‴,6‴), 128.3 (C-8′), 128.1 (C-1′), 127.7 (C-4″), 127.5 (C-2″,6″), 126.9 (C-9′a), 126.6 (C-8′a), 126.2 (C-3′), 126.0 (C-7′), 125.7 (C-2′), 89.1 (C-5), 63.1 (C-4) ppm.

4-[3-(4-Bromophenyl)-4-phenyl-4,5-dihydro-1,2-oxazole-5-yl]acridine (**8c**). Yield: 39.2 mg (23.0%). Mp. 185–187 °C. Yellow powder. For C_28_H_19_BrN_2_O (479.38) found: C 70.03, H 3.88, N 5.71%; calc.: C 70.16, H 4.00, Br 16.67, N 5.84, O 3.34%. *R_f_* (5:1 *v*/*v c*-Hex/EtOAc) 0.46. FT-IR: ν_max_ 3105, 2920, 1634, 1330, 1060, 1025, 950, 917, 758 cm^−1^. ^1^H NMR (400 MHz CDCl_3_): δ 8.79 (1H, s, H-9′), 8.09 (1H, d, *J* = 8.8 Hz, H-5′), 8.02 (1H, d, *J* = 8.4 Hz, H-8′), 7.94 (2H, m, H-1′, H-3′), 7.79 (1H, ddd, *J* = 8.8, 6.6, 1.4 Hz, H-6′), 7.58 (2H, d, *J* = 8.0 Hz, H-2″,6″), 7.53 (1H, m, H-2′, H-7′), 7.49 (4H, m, H-3″,5″, H-2‴,6‴), 7.40 (1H, t, *J* = 7.3 Hz, H-4″), 7.34 (2H, d, *J* = 8.7 Hz, H-3‴,5‴), 6.71 (1H, d, *J* = 3.5 Hz, H-5), 4.84 (1H, d, *J* = 3.5 Hz, H-4) ppm. ^13^C NMR (100.6 MHz, CDCl_3_): δ 158.3 (C-3), 148.3 (C-10′a), 146.4 (C-4′a), 139.4 (C-1″), 138.0 (C-4′), 136.2 (C-9′), 131.9 (C-3‴,5‴), 130.4 (C-6′), 129.7 (C-5′), 129.2 (C-3″,5″), 128.9 (C-2‴,6‴), 128.4 (C-2″,6″), 128.3 (C-8′), 128.2 (C-1‴, C-1′), 127.9 (C-4″), 126.9 (C-8′a), 126.6 (C-9′a), 126.2(C-3′), 126.1 (C-7′), 125.6 (C-2′), 124.2 (C-4‴), 89.4 (C-5), 62.9 (C-4) ppm.

4-[3-(3-Nitrophenyl)-4-phenyl-4,5-dihydro-1,2-oxazole-5-yl]acridine (**8d**). The compound **8e** was obtained as a mixture of **7d** and **8d**. *R_f_* (5:1 *v*/*v c*-Hex/EtOAc) 0.32. ^1^H NMR (600 MHz CDCl_3_): δ 8.81 (1H, s, H-9′), 8.41 (1H, t, *J* = 2.0 Hz, H-2‴), 8.09 (1H, ddd, *J* = 8.2, 2.3, 1.1 Hz, H-4‴), 8.06 (1H, m, H-5′), 8.03 (1H, ddt, *J* = 8.4, 1.4, 0.6 Hz, H-8′), 7.97 (1H, m, H-1′), 7.96 (1H, ddd, *J* = 8.0, 1.7, 1.1 Hz, H-6‴), 7.94 (dt, *J* = 6.9, 1.3 Hz, H-3′), 7.79 (1H, ddd, *J* = 8.8, 6.6, 1.4 Hz, H-6′), 7.61 (2H, m, H-2″,6″), 7.58 (1H, ddd, *J* = 8.4, 6.6, 1.1 Hz, H-7′), 7.54 (1H, dd, *J* = 8.4, 6.9 Hz, H-2′), 7.51 (2H, m, H-3″,5″), 7.42 (1H, m, H-4″), 7.41 (2H, t, *J* = 8.0 Hz, H-3‴,5‴) ppm. ^13^C NMR (150.1 MHz, CDCl_3_): δ 157.5 (C-3), 148.3 (C-3‴), 148.2 (C-10′a), 146.2 (C-4′a), 138.7 (C-1″), 137.5 (C-4′), 136.1 (C-9′), 132.8 (C-6‴), 131.0 (C-1‴), 130.4 (C-6′), 129.6 (C-5‴), 129.5 (C-5′), 129.2 (C-3″,5″), 128.6 (C-1′), 128.3 (C-2″,6″), 128.2 (C-8′), 128.1 (C-4″), 126.8 (C-9′a), 126.5 (C-8′a), 126.0 (C-7′), 125.9 (C-3′), 125.4 (C-2′), 124.2 (C-4‴), 122.1 (C-2‴), 89.9 (C-5), 62.5 (C-4) ppm.

4-[3-(4-Nitrophenyl)-4-phenyl-4,5-dihydro-1,2-oxazole-5-yl]acridine (**8e**). The compound 8**d** was obtained as a mixture of **7e** and **8e**. *R_f_* (5:1 *v*/*v c*-Hex/EtOAc) 0.32. ^1^H NMR (400 MHz CDCl_3_): δ 8.78 (1H, s, H-9′), 8.06 (2H, m, H-3‴,5‴), 8.04 (1H, m, H-5′), 8.01 (1H, m, H-8′), 7.96 (1H, dd, *J* = 8.4, 0.8 Hz, H-1′), 7.91 (1H, dt, *J* = 6.9, 1.3 Hz, H-2′), 7.78 (1H, ddd, *J* = 8.8, 6.7, 1.5 Hz, H-6′), 7.76 (2H, d, *J* = 9.2 Hz, H-2‴,6‴), 7.62 (1H, m, H-7′), 7.56 (2H, m, H-2″,6″), 7.53 (1H, m, H-2′), 7.50 (2H, m, H-3″,5″), 7.41 (1H, m, H-4″), 6.76 (1H, d, *J* = 3.7 Hz, H-5), 4.91 (1H, d, *J* = 3.7 Hz, H-4) ppm. ^13^C NMR (100.6 MHz, CDCl_3_): δ 157.6 (C-3), 148.2 (C-10′a), 148.1 (C-4‴), 146.2 (C-4′a), 138.8 (C-1″), 137.4 (C-4′), 136.2 (C-9′), 135.3 (C-1‴), 130.4 (C-6′), 129.5 (C-5′), 128.6 (C-3″,5″), 128.3 (C-1′), 128.2 (C-2″,6″), 128.1 (C-4″, C-8′, C-2‴,6‴), 126.8 (C-9′a), 126.5 (C-8′a), 126.0 (C-3′, C-7′), 125.6 (C-2′), 123.8 (C-3‴,5‴), 90.2 (C-5), 62.0 (C-4) ppm.

### 3.8. Synthesis of 5-(Acridin-4-yl)-3-phenyl-4,5-dihydro-1,2-oxazole-4-carboxylic Acids ***9a**,**b**,**d**,**e***, (4Z)-4-[(Acridin-4-yl)Methylidene)]-3-(4-Nitrophenyl)-4,5-Dihydro-1,2-Oxazol-5-One Z-***10e*** and (4E)-4-[(Acridin-4-yl)Methylidene)]-3-(4-Nitrophenyl)-4,5-Dihydro-1,2-Oxazol-5-One E-***10e***

To a solution of methylester (**6a**: 100 mg; **6b**: 100 mg, 0.261 mmol, 0.24 mmol; **6d**: 100 mg, 0.23 mmol; **6e**: 100 mg, 0.23 mmol) in ethanol (5 mL) heated to 40 °C, KOH (**6a**: 136 mg, 2.40 mmol; **6b**: 146 mg, 2.61 mmol; **6e**: 146 mg, 2.30 mmol; **6d**: 146 mg, 2.30 mmol) was added. The reaction mixture was stirred at 60 °C and monitored using TLC (SiO_2_, *c*-Hex/EtOAc, 1:1). Upon complexion of the reaction, the solvent was evaporated, and water (10 mL) was added. The resulting aqueous solution was acidified (HCl, 3:1), and the precipitate was extracted with diethyl ether (2 × 10 mL). The organic layer was dried, and the solvent was subsequently evaporated to yield the crude products. These products were further purified by column chromatography (SiO_2_, *c*-Hex/EtOAc, 1:1). 

The compounds *Z*-**10e** and *E*-**10e** were separated from the mixture of **6e** and **9e** using column chromatography (SiO_2_, CHCl_3_/MeOH, 4:1). The ratio of isomers *Z*-**10e** and *E*-**10e** was determined to be 1.00:0.14 based on ^1^H NMR spectra.

5-(Acridin-4-yl)-3-(4-methoxyphenyl)-4,5-dihydro-1,2-oxazole-4-carboxylic acid (**9a**). Yield: 76.3 mg (79.0%). Mp. 190–192 °C. Yellow powder. For C_24_H_18_N_2_O_4_ (398.42) found: C 72.15, H 4.75, N 7.14%; calc.: C 72.35, H 4.55, N 7.03, O 16.06%. *R_f_* (*c*-Hex/EtOAc, 1:1) 0.31. FT-IR: ν_max_ 3443, 3117, 2929, 1788, 1643, 1622, 1525, 1353, 1243, 1209, 1115, 1066, 1038, 711, 617 cm^−1^. ^1^H NMR (600 MHz CDCl_3_): δ 9.05 (1H, s, H-9′), 8.40 (1H, d, *J* = 8.4 Hz, H-5′), 8.16 (1H, dt, *J* = 7.3, 1.5 Hz, H-3′), 8.13 (1H, d, *J* = 8.6 Hz, H-8′), 8.06 (1H, d, *J* = 8.4 Hz, H-1′), 7.96 (1H, ddd, *J* = 8.4, 6.7, 1.4 Hz, H-6′), 7.70 (1H, ddd, *J* = 8.6, 6.7, 1.0 Hz, H-7′), 7.64 (1H, m, H-2′), 7.62 (2H, d, *J* = 9.1 Hz, H-2″,6″), 6.85 (2H, d, *J* = 9.1 Hz, H-3″,5″), 6.83 (1H, d, *J* = 5.5 Hz, H-5), 4.76 (1H, d, *J* = 5.5 Hz, H-4), 3.79 (3H, s, OCH_3_) ppm. ^13^C NMR (150.1 MHz, CDCl_3_): δ 169.5 (C-6), 161.3 (C-4″), 153.8 (C-3), 147.1 (C-10′a), 144.2 (C-4′a), 140.0 (C-9′), 135.6 (C-4′), 132.9 (C-6′), 129.0 (C-2″,6″), 128.8 (C-1′), 128.7 (C-8′), 127.7 (C-3′), 127.3 (C-9′a), 127.0 (C-7′), 126.8 (C-8′a), 126.5 (C-5′), 126.1 (C-2′), 121.5 (C-1″), 114.2 (C-3″,5″), 82.3 (C-5), 62.4 (C-4), 55.3 (OCH_3_) ppm.

5-(Acridin-4-yl)-3-phenyl-4,5-dihydro-1,2-oxazole-4-carboxylic acid (**9b**). Yield: 94.3 mg (84.0%). Mp. 190–192 °C. Yellow powder. For C_23_H_16_N_2_O_3_ (368.40) found: C 74.85, H 4.22, N 7.68%; calc.: C 74.99, H 4.38, N 7.60, O 13.03%; *R_f_* (*c*-Hex/EtOAc, 1:1) 0.34. FT-IR: ν_max_ 3428, 3112, 2906, 1799, 1231, 1198, 1040, 508 cm^−1^. ^1^H NMR (600 MHz CDCl_3_): δ 9.06 (1H, s, H-9′), 8.41 (1H, d, *J* = 8.4 Hz, H-5′), 8.16 (1H, dd, *J* = 7.0, 1.4 Hz, H-3′), 8.14 (1H, d, *J* = 8.0 Hz, H-8′), 8.07 (1H, d, *J* = 8.4 Hz, H-1′), 7.97 (1H, ddd, *J* = 8.4, 6.8, 1.5 Hz, H-6′), 7.70 (3H, m, H-7′, H-2″,6″), 7.65 (1H, dd, *J* = 8.4, 7.0 Hz, H-2′), 7.35 (3H, m, H-3″,5″, H-4″), 6.87 (1H, d, *J* = 5.5 Hz, H-5), 4.80 (1H, d, *J* = 5.5 Hz, H-4) ppm. ^13^C NMR (150.1 MHz, CDCl_3_): δ 169.4 (C-6), 154.3 (C-3), 147.0 (C-10′a), 144.2 (C-4′a), 140.0 (C-9′), 135.4 (C-4′), 132.8 (C-6′), 130.4 (C-4″), 128.9 (C-1′, C-8′), 128.8 (C-3″,5″), 128.4 (C-1″), 127.7 (C-3′), 127.5 (C-2″,6″), 127.1 (C-7′), 126.9 (C-8′a), 126.6 (C-9′a), 126.5 (C-5′), 126.2 (C-2′), 82.6 (C-5), 62.1 (C-4) ppm.

5-(Acridin-4-yl)-3-(3-nitrophenyl)-4,5-dihydro-1,2-oxazole-4-carboxylic acid (**9d**). Yield: 95.7 mg (99.0%). Mp. 183–185 °C. For C_23_H_15_N_3_O_5_ (413.39) found: C 66.71, H 3.56, N 10.22%; calc.: C 66.83, H 3.66, N 10.16, O 19.35%. *R_f_* (1:1 *v*/*v c*-Hex/EtOAc) 0.11. FT-IR: ν_max_ 3445, 3107, 1792, 1659, 1634, 1354, 1149, 1136, 1082, 1054, 733, 695, 617 cm^−1^. ^1^H NMR (600 MHz CDCl_3_): δ 9.09 (1H, s, H-9′), 8.45 (1H, dd, *J* = 2.1, 1.7 Hz, H-2″), 8.43 (1H, dd, *J* = 8.8, 1.0 Hz, H-5′), 8.22 (1H, ddd, *J* = 8.2, 2.2, 1.1 Hz, H-4″), 8.16 (2H, m, H-3′, H-3′), 8.11 (2H, m, H-1′, H-6″), 8.00 (1H, ddd, *J* = 8.8, 6.6, 1.4 Hz, H-6′), 7.73 (1H, ddd, *J* = 8.4, 6.6, 1.0 Hz, H-7′), 7.67 (1H, dd, *J* = 8.4, 7.0 Hz, H-2′), 7.55 (1H, m, H-5″), 6.94 (1H, d, *J* = 5.9 Hz, H-5), 4.85 (1H, d, *J* = 5.9 Hz, H-4) ppm. ^13^C NMR (150.1 MHz, CDCl_3_): δ 169.4 (C-4), 154.3 (C-3), 148.3 (C-3″), 147.0 (C-10′a), 144.2 (C-4′a), 140.0 (C-9′), 135.4 (C-4′), 133.0 (C-6″), 132.8 (C-6′), 130.9 (C-1″), 128.9 (C-1′, C-8′), 128.8 (C-5″), 127.6 (C-3′), 127.1 (C-7′), 126.9 (C-8′a), 126.6 (C-9′a), 126.5 (C-5′), 126.2 (C-2′), 124.6 (C-4″), 122.2 (C-2″), 82.6 (C-5), 62.1 (C-4) ppm.

5-(Acridin-4-yl)-3-(4-nitrophenyl)-4,5-dihydro-1,2-oxazole-4-carboxylic acid (**9e**). Yield: 58.7 mg (63.0%). Mp. 250–253 °C. Yellow powder. For C_23_H_15_N_3_O_5_ (413.39) found: C 66.79, H 3.45, N 10.14%; calc.: C 66.83, H 3.66, N 10.17, O 19.35%. *R_f_* (*c*-Hex/EtOAc, 1:1) 0.16. FT-IR: ν_max_ 3486, 3102, 2946, 2916, 1798, 1655, 1570, 1333, 1205, 1171, 1080, 1057, 831, 460 cm^−1^. ^1^H NMR (600 MHz CDCl_3_): δ 9.09 (1H, s, H-9′), 8.42 (1H, d, *J* = 8.4 Hz, H-5′), 8.21 (2H, d, *J* = 9.2 Hz, H-3″,5″), 8.16 (2H, m, H-3′, H-8′), 8.11 (1H, dd, *J* = 8.4, 1.4 Hz, H-1′), 7.99 (1H, ddd, *J* = 8.4, 6.4, 1.2 Hz, H-6′), 7.87 (2H, d, *J* = 9.1 Hz, H-2″,6″), 7.73 (1H, ddd, *J* = 8.4, 6.4, 1.2 Hz, H-7′), 7.67 (1H, t, *J* = 7.7 Hz, H-2′), 6.94 (1H, d, *J* = 6.0 Hz, H-5), 4.82 (1H, d, *J* = 6.0 Hz, H-4) ppm. ^13^C NMR (150.1 MHz, CDCl_3_): δ 167.8 (C-6), 140.2 (C-9′), 133.2 (C-6′), 129.1 (C-1′), 128.6 (C-8′), 128.2 (C-2″,6″), 127.6 (C-3′), 127.1 (C-7′), 126.0 (C-5′), 125.9 (C-2′), 123.8 (C-3″,5″), 83.3 (C-5), 61.4 (C-4) ppm. The resonance lines for carbons C-3, C-4′, C-4′a, C-10′a, C-8′a, C-9′a, C-1″, C-4″ were not detected.

(4*Z*)-4-[(Acridin-4-ylmethylidene)]-3-(4-nitrophenyl)-4,5-dihydro-1,2-oxazole-5-one (***Z*-10e**). The compound was obtained as a mixture of *Z*-**10e** and *E*-**10e**. *R_f_* (4:1 *v*/*v* CHCl_3_/MeOH) 0.80. ^1^H NMR (600 MHz CDCl_3_): δ 9.85 (1H, s, H-6), 9.73 (1H, dd, *J* = 7.3, 1.4 Hz, H-3′), 8.87 (1H, s, H-9′), 8.54 (2H, d, *J* = 8.9 Hz, H-3″,5″), 8.33 (1H, dd, *J* = 8.4, 0.6 Hz, H-1′), 8.08 (2H, d, *J* = 8.9 Hz, H-2″,6″), 8.08 (1H, m, H-5′), 8.06 (1H, dd, *J* = 8.4, 1.2 Hz, H-8′), 7.86 (1H, ddd, *J* = 8.4, 6.6, 1.2 Hz, H-6′), 7.77 (1H, dd, *J* = 8.4, 7.3 Hz, H-2′), 7.63 (1H, ddd, *J* = 8.4, 6.6, 1.2 Hz, H-7′) ppm. ^13^C NMR (150.1 MHz, CDCl_3_): δ 168.5 (C-5), 162.7 (C-3), 149.6 (C-4″), 149.3 (C-6), 148.8 (C-10′a), 147.0 (C-4′a), 137.3 (C-9′), 137.2 (C-3′), 136.0 (C-1′), 134.1 (C-1″), 131.6 (C-6′), 130.1 (C-2″,6″), 129.7 (C-5′), 129.3 (C-4′), 128.2 (C-8′), 126.9 (C-8′a), 126.8 (C-7′), 126.2 (C-9′a), 125.6 (C-2′), 124.4 (C-3″,5″), 121.4 (C-4) ppm.

### 3.9. Synthesis of 4-(3-Phenyl-4,5-dihydro-1,2-oxazol-5-yl)acridines ***11a**,**b**,**d**,**e***

Carboxylic acids **9a**,**b**,**d**,**e** (20 mg) were dissolved in deuterated chloroform (0.6 mL) and allowed to stand at room temperature for 3 weeks. The reactions were monitored by ^1^H NMR spectroscopy.

4-[3-(4-Methoxyphenyl)-4,5-dihydro-1,2-oxazol-5-yl]acridine (**11a**). The compound was obtained as a mixture of **9a** and **11a**. Yield: 61.0% (from ^1^H NMR). *R_f_* (3:1 *v*/*v* toluene/EtOAc) 0.68. ^1^H NMR (600 MHz CDCl_3_): δ 8.77 (1H, s, H-9′), 8.23 (1H, d, *J* = 8.4 Hz, H-5′), 8.01 (1H, dd, *J* = 8.4, 1.5 Hz, H-8′), 7.98 (1H, dd, *J* = 6.9, 1.3 Hz, H-3′), 7.94 (1H, dd, *J* = 8.4, 1.4 Hz, H-1′), 7.79 (1H, ddd, *J* = 8.4, 6.6, 1.4 Hz, H-6′), 7.65 (2H, d, *J* = 9.0 Hz, H-2″,6″), 7.56 (1H, ddd, *J* = 8.4, 6.6, 1.1 Hz, H-7′), 7.53 (1H, dd, *J* = 8.4, 6.9 Hz, H-2′), 6.89 (2H, d, *J* = 9.0 Hz, H-3″,5″), 6.89 (1H, m, H-5), 4.24 (1H, dd, *J* = 17.0, 11.2 Hz, H-4a), 3.81 (3H, s, OCH_3_), 3.36 (1H, dd, *J* = 17.0, 7.5 Hz, H-4b) ppm. ^13^C NMR (150.1 MHz, CDCl_3_): δ 161.0 (C-4″), 156.6 (C-3), 148.2 (C-10′a), 146.2 (C-4′a), 139.2 (C-4′), 136.0 (C-9′), 130.1 (C-6′), 129.9 (C-5′), 128.3 (C-2″,6″), 128.1 (C-8′), 127.8 (C-1′), 126.6 (C-9′a), 126.4 (C-8′a), 126.0 (C-3′), 125.9 (C-7′), 125.6 (C-2′), 122.4 (C-1″), 114.1 (C-3″,5″), 79.5 (C-5), 55.3 (OCH_3_), 43.9 (C-4) ppm.

4-(3-Phenyl-4,5-dihydro-1,2-oxazol-5-yl)acridine (**11b**). The compound was obtained as a mixture of **9b** and **11b**. Yield: 94.0% (from ^1^H NMR). *R_f_* (5:1 *v*/*v n*-Hex/EtOAc) 0.29. ^1^H NMR (400 MHz CDCl_3_): δ 8.78 (1H, s, H-9′), 8.23 (1H, d, *J* = 8.4 Hz, H-5′), 8.02 (1H, d, *J* = 8.4 Hz, H-8′), 7.98 (1H, dd, *J* = 6.9, 1.3 Hz, H-3′), 7.95 (1H, d, *J* = 8.4 Hz, H-1′), 7.79 (1H, ddd, *J* = 8.4, 6.6, 1.4 Hz, H-6′), 7.72 (2H, m, H-2″,6″), 7.55 (2H, m, H-2′, H-7′), 7.38 (3H, m, H-3″,5″, H-4″), 6.93 (1H, dd, *J* = 11.2, 7.5 Hz, H-5), 4.28 (1H, dd, *J* = 17.1, 11.2 Hz, H-4a), 3.41 (1H, dd, *J* = 17.1, 7.5 Hz, H-4b) ppm. ^13^C NMR (100.6 MHz, CDCl_3_): δ 158.1 (C-3), 149.3 (C-10′a), 147.2 (C-4′a), 140.1 (C-4′), 137.0 (C-9′), 131.2 (C-6′), 131.0 (C-5′, C-4″), 129.7 (C-1″, C-3″,5″), 129.1 (C-8′), 128.9 (C-1′), 127.8 (C-2″,6″), 127.7 (C-9′a), 127.5 (C-8′a), 126.9 (C-3′, C-7′), 126.6 (C-2′), 80.9 (C-5), 44.7 (C-4) ppm.

4-[3-(3-Nitrophenyl)-4,5-dihydro-1,2-oxazol-5-yl]acridine (**11d**). Yield: 50.0% (from ^1^H NMR). *R_f_* (1:1 *v*/*v c*-Hex/EtOAc) 0.82. ^1^H NMR (600 MHz, CDCl_3_): δ 8.80 (1H, s, H-9′), 8.46 (1H, dd, *J* = 2.3, 1.2 Hz, H-2″), 8.24 (1H, ddd, *J* = 8.2, 2.3, 1.1 Hz, H-4″), 8.22 (1H, d, *J* = 8.4 Hz, H-5′), 8.14 (1H, m, H-6″), 8.03 (1H, dd, *J* = 8.4, 1.4 Hz, H-8′), 7.98 (1H, m, H-1′), 7.81 (1H, ddd, *J* = 8.4, 6.6, 1.4 Hz, H-6′), 7.59 (1H, m, H-7′, H-5″), 7.55 (1H, m, H-2′), 7.56 (2H, m, H-2′,), 6.98 (1H, dd, *J* = 11.5, 7.9 Hz, H-5), 4.32 (1H, dd, *J* = 17.1, 11.4 Hz, H-4a), 3.46 (1H, dd, *J* = 17.1, 7.8 Hz, H-4b) ppm. ^13^C NMR (150.1 MHz, CDCl_3_): δ 155.5 (C-3), 148.5 (C-3″), 148.3 (C-10′a), 146.1 (C-4′a), 138.4 (C-4′), 136.1 (C-9′), 132.3 (C-6″), 131.8 (C-1″), 130.3 (C-6′), 129.9 (C-5′), 129.7 (C-5″), 128.2 (C-8′), 128.1 (C-1′), 126.8 (C-9′a), 126.5 (C-8′a), 126.0 (C-3′, C-7′), 125.4 (C-2′), 124.4 (C-4″), 121.6 (C-2″), 80.9 (C-5), 43.2 (C-4) ppm.

4-[3-(4-Nitrophenyl)-4,5-dihydro-1,2-oxazol-5-yl]acridine (**11e**). Yield: 36.0% (from ^1^H NMR). *R_f_* (1:1 *v*/*v c*-Hex/EtOAc) 0.83. ^1^H NMR (600 MHz CDCl_3_): δ 8.80 (1H, s, H-9′), 8.25 (1H, d, *J* = 9.0 Hz, H-5″), 8.20 (1H, dd, *J* = 8.7, 1.0 Hz, H-5′), 8.03 (1H, dd, *J* = 8.4, 1.4 Hz, H-8′), 7.98 (1H, dd, *J* = 8.4, 1.4 Hz, H-1′), 7.96 (1H, dd, *J* = 6.8, 1.3 Hz, H-3′), 7.89 (2H, d, *J* = 9.0 Hz, H-2″,6″), 7.80 (1H, ddd, *J* = 8.7, 6.6, 1.4 Hz, H-6′), 7.58 (1H, ddd, *J* = 8.4, 6.6, 1.2 Hz, H-7′), 7.55 (1H, dd, *J* = 8.4, 7.2 Hz, H-2′), 6.97 (1H, dd, *J* = 11.5, 7.9 Hz, H-5), 4.30 (1H, dd, *J* = 17.1, 11.5 Hz, H-4a), 3.45 (1H, dd, *J* = 17.1, 7.9 Hz, H-4b) ppm. ^13^C NMR (150.1 MHz, CDCl_3_): δ 155.7 (C-3), 148.4 (C-4″), 148.3 (C-10′a), 146.1 (C-4′a), 138.3 (C-4′), 136.1 (C-9′, C-1″), 130.3 (C-6′), 129.9 (C-5′), 128.2 (C-1′, C-8′), 127.5 (C-2″,6″), 126.7 (C-8′a), 126.6 (C-9′a), 126.0 (C-3′), 125.9 (C-7′), 125.4 (C-2′), 124.0 (C-3″,5″), 81.2 (C-5), 43.0 (C-4) ppm.

## 4. Conclusions

This study centred on the investigation of [3 + 2] cycloaddition reactions involving acridine-alkenes, specifically methyl-(*2E*)-3-(acridin-4-yl)-prop-2-enoate (**1**) and 4-[(1*E*)-2-phenylethenyl]acridine (**2**), with unstable benzonitrile N-oxides **4a**–**e**. The reaction exhibited distinct regioselectivity patterns influenced by the polarity of the C=C double bond of the dipolarophile, electronic factors of the dipole, and steric factors.

In the case of alkene **1**, the regioisomers **5** and **6** were formed favouring the 5-(acridin-4-yl)-4,5-dihydro-1,2-oxazole-4-carboxylates **6a**,**b**,**d**,**e**. Remarkably, the reaction with 4-methoxybenzonitrile oxide (**4a**) exclusively produced the regioisomeric derivative **6a**. In contrast, the regioselectivity was reversed for the reactions of alkene **2** with nitrile oxides **4c**–**e**, favouring the formation of products **7c**–**e**. An interesting regioselectivity pattern was observed in the reaction with **4a**, yielding major isoxazoline **8a**.

The subsequent hydrolysis of isolated esters **6a**,**b**,**d**,**e** resulted in the production of carboxylic acids **9a**,**b**,**d**,**e** with nearly 100% conversion. Hydrolysis of derivative **6e** led to the formation of three products: acid **9e** (81%) and two isoxazole-5-one stereoisomers, *Z*-**10e** (14%) and *E*-**10e** (5%). However, the separation and purification of isoxazole-5-ones *Z*-**10e** and *E*-**10e** proved challenging, yielding mainly a mixture of isoxazole-5-one *Z*-**10e** (87%) along with the *E*-**10e** (13%) derivative after repeated crystallization.

Notably, during NMR measurements of carboxylic acids **9a**,**b**,**d**,**e** in CDCl_3_, decarboxylation was observed, manifesting in NMR spectra featuring two distinct sets of signals. The appearance of three new signals corresponding to protons H-4a, H-4b, and H-5, along with non-equivalent signals for protons H-4a and H-4b, indicated the formation of a new prochiral carbon centre C-4. The occurrence of decarboxylation was further conformed by a noticeable colour change from light yellow to dark green. 

In conclusion, this comprehensive investigation provides valuable insights into the regioselectivity of [3 + 2] cycloadditions, and the subsequent transformations of the synthesised compounds, paving the way for further exploration of their potential applications in diverse scientific domains.

## Data Availability

The data presented in this study and associated additional data are available upon request.

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
