# Peer review of "Spectral Assignment in the [3 + 2] Cycloadditions of Methyl (2E)-3-(Acridin-4-yl)-prop-2-enoate and 4-[(E)-2-Phenylethenyl]acridin with Unstable Nitrile N-Oxides"

_molecules, 2024, doi:10.3390/molecules29122756_

Round 1

Reviewer 1 Report

Comments and Suggestions for Authors

The manuscript "Spectral Assignment in the 1,3-Dipolar Cycloadditions of Methyl (2E)-3-(acridin-4-yl)-prop-2-enoate and 4-[(E)-2-Phenylethenyl]acridin with Unstable Nitrile Oxides" is devoted to the synthesis of a hoard of 1,2-oxazole derivatives via 1,3 dipolar cycloaddition reaction. The known issue of the synthesis (low stability of aromatic nitrile oxides) was elegantly circumvented by the Authors - they allowed nitrile oxides to be generated in situ. The synthesized compounds were thoroughly characterized using different 1D and 2D NMR techniques, which is great. Some regioisomeric products were inseparable using column chromatography despite the Authors' best efforts. However, the Authors were able to assign the chemical shifts in the spectra of mixtures. In general, the paper is of great interest to organic chemists who specialize in cycloaddition reactions and to a broader audience as a good example of NMR spectra assignment.

The only minor comment: the structures 3a-e should also be given in the text explicitly.

Author Response

Dear Reviewer 1,

I hope this letter finds you in good health and spirits. I am writing to provide an update on our manuscript entitled “Spectral Assignment in the [3+2] Cycloadditions of Methyl (2E)-3-(acridin-4-yl)-prop-2-enoate and 4-[(E)-2-Phenylethenyl]acridin with Unstable Nitrile N-Oxides” to the special issue New Insights into Nuclear Magnetic Resonance (NMR) Spectroscopy for consideration for publication in Molecules.

We are pleased to inform you that we have carefully addressed all the suggestions and comments. Thank you for your valuable feedback. The structure of derivatives 3a-e was added to Scheme 1.

Sincerely,

Assoc. prof. Mária Vilková

Reviewer 2 Report

Comments and Suggestions for Authors

The manuscript "Spectral Assignment in the 1,3-Dipolar Cycloadditions of Me-thyl (2E)-3-(acridin-4-yl)-prop-2-enoate and 4-[(E)-2-Phe-nylethenyl]acridin with Unstable Nitrile Oxides" containt results of the experimental study regarding to interesting group of the cycloaddition processes. The topic is valuable, but the work include important weak points, which must be corrected before the further evaluation. In parcitular:

According to the actual state of knowledge, almost all compounds defined earlier as "1,3-dipole" exhibit not a dipolar nature [Eur. J. Org. Chem. 267–282 (2019)]. In the consequence, term "1,3-dipolar cycloaddition" should be replaced to the "[3+2] cycloaddition"; "1,3-dipole" to "thre atom 4pi-component", "dipolarophile" to "2pi-component" etc consequently.

Page 1, sentence: "On the other hand, the reaction of nitrile oxides with 1,2-disubstituted alkenes tends to be slower and usually afford mixtures of regio- and stereoisomers."
This is not true. This is observed only in the reaction with the participation of non-activated alkenes. In the case of activated alkenes such as nitroalkenes, full regioselective [3+2] cycloadditions are possible [Molecules, 26, 6774 (2021)]. So, this sentence should be modified according to actual state of knowledge.

Page 1, sentence: "Computational studies of 1,3-dipolar cycloadditions have been carried out to rationalize the reactivity and regioselectivity of these reactions.[15]"
This reference is adequate only for cycloadditions between Nitrile Oxides and α,β-Unsaturated Amides. For this type, general conclusion, a wider range of sources from different research groups basen on the recent literature should be provided (for example: Organics, 2, 26 (2021); Scientiae Radices, 2, 75-92 (2023); Chem Heterocycl Comp 59, 145–154 (2023);  Chem Heterocycl Comp 59, 155–164 (2023)).
Next, the ref.15 is based on the outdated FMO approach. At this moment, the analysis of the distribution of local electrophilicities and nucleophilicities is dedicated for the interpretation of the regioselectivity of the [3+2] cycloaddition [Molecules 21, 748 (2016)].

Scheme 1 and discussion in the text:
Reaction components and product should be presented in the order depend of the Hammet constants of substituents. Next, values of respective Hammett constants should be added to the table. This will be helpfull for the analysis of the nitrile N-oxide reactivity.
Next, the stage of the formation of nitrile oxide should be included to the Scheme.

The question of the stability and tendency for the dimerization of nitrile oxides were full explored by Grundmann in the range 1965-1070. So, respective references from this area should be cited instead of the ref.22.

Compounds metrics (including data from the NMR, IR and HRMS analysis as well as m.p's and informations about the yields) should be collected in the main file of the manuscript.

Lastly, some technical errors (sentences such as "Error! Reference source not found.") should be corrected along to all parte of the manuscript.

Author Response

Dear Reviewer 2,

I hope this letter finds you in good health and spirits. I am writing to provide an update on our manuscript entitled “Spectral Assignment in the [3+2] Cycloadditions of Methyl (2E)-3-(acridin-4-yl)-prop-2-enoate and 4-[(E)-2-Phenylethenyl]acridin with Unstable Nitrile N-Oxides” to the special issue New Insights into Nuclear Magnetic Resonance (NMR) Spectroscopy for consideration for publication in Molecules.

We are pleased to inform you that we have carefully addressed all the suggestions and comments.

We are sincerely grateful for your thorough comments, valuable insights, and the time you dedicated to reviewing our manuscript. Your feedback is instrumental in improving the quality and clarity of our work. We will carefully consider all your suggestions and make the necessary corrections to ensure the manuscript meets the highest standards.

Thank you once again for your time and effort in evaluating our research.

Our responses:

The manuscript "Spectral Assignment in the 1,3-Dipolar Cycloadditions of Me-thyl (2E)-3-(acridin-4-yl)-prop-2-enoate and 4-[(E)-2-Phe-nylethenyl]acridin with Unstable Nitrile Oxides" containt results of the experimental study regarding to interesting group of the cycloaddition processes. The topic is valuable, but the work include important weak points, which must be corrected before the further evaluation. In parcitular: According to the actual state of knowledge, almost all compounds defined earlier as "1,3-dipole" exhibit not a dipolar nature [Eur. J. Org. Chem. 267–282 (2019)]. In the consequence, term "1,3-dipolar cycloaddition" should be replaced to the "[3+2] cycloaddition"; "1,3-dipole" to "thre atom 4pi-component", "dipolarophile" to "2pi-component" etc consequently.

We appreciate your comments and have made the necessary corrections as per your suggestions. We have updated the terminology throughout our manuscript. These changes have been consistently applied throughout the manuscript to align with the current state of knowledge and to avoid any terminological ambiguities. We believe these corrections address the important weak points you highlighted, and we are confident that the revised manuscript now provides a clearer and more accurate representation of our experimental study.

Page 1, sentence: "On the other hand, the reaction of nitrile oxides with 1,2-disubstituted alkenes tends to be slower and usually afford mixtures of regio- and stereoisomers." This is not true. This is observed only in the reaction with the participation of non-activated alkenes. In the case of activated alkenes such as nitroalkenes, full regioselective [3+2] cycloadditions are possible [Molecules, 26, 6774 (2021)]. So, this sentence should be modified according to actual state of knowledge.

We acknowledge your point regarding the reactivity of activated versus non-activated alkenes. The sentence has been revised.

Page 1, sentence: "Computational studies of 1,3-dipolar cycloadditions have been carried out to rationalize the reactivity and regioselectivity of these reactions.[15]" This reference is adequate only for cycloadditions between Nitrile Oxides and α,β-Unsaturated Amides. For this type, general conclusion, a wider range of sources from different research groups basen on the recent literature should be provided (for example: Organics, 2, 26 (2021); Scientiae Radices, 2, 75-92 (2023); Chem Heterocycl Comp 59, 145–154 (2023);  Chem Heterocycl Comp 59, 155–164 (2023)). Next, the ref.15 is based on the outdated FMO approach. At this moment, the analysis of the distribution of local electrophilicities and nucleophilicities is dedicated for the interpretation of the regioselectivity of the [3+2] cycloaddition [Molecules 21, 748 (2016)].

The sentence on Page 1 about computational studies of 1,3-dipolar cycloadditions has been revised to include a wider range of sources from different research groups, as well as a more current approach focusing on the distribution of local electrophilicities and nucleophilicities for interpreting regioselectivity.

Scheme 1 and discussion in the text:

Reaction components and product should be presented in the order depend of the Hammet constants of substituents. Next, values of respective Hammett constants should be added to the table. This will be helpfull for the analysis of the nitrile N-oxide reactivity.

We have rearranged the nitrile oxides used in the study according to their respective Hammett constants. However, we have some reservations about discussing the reactivity of nitrile oxides purely in terms of their Hammett constants. Given that our experimental procedure involved the slow addition of TEA over 8 days to prevent the formation of furoxan, it is challenging to accurately assess the reactivity of nitrile oxides towards alkenes under these specific conditions. Therefore, we have not included the values of the relevant Hammett constants in the table.

Next, the stage of the formation of nitrile oxide should be included to the Scheme.

We have updated Scheme 1 to include the stage of the formation of the nitrile oxide. This addition provides a more comprehensive overview of the reaction process and clarifies the sequence of events.

The question of the stability and tendency for the dimerization of nitrile oxides were full explored by Grundmann in the range 1965-1070. So, respective references from this area should be cited instead of the ref.22.

We have updated our manuscript to include references that more comprehensively explore the stability and tendency for the dimerization of nitrile oxides, as originally examined by Grundmann. Unfortunately, we do not have access to Grundmann's original works. Instead, we have included the following relevant references:

Barbaro, G.; Battaglia, A.; Dondoni, A. Kinetics and Mechanism of Dimerisation of Benzonitrile N-Oxides to Furazan N-Oxides. J. Chem. Soc. B Phys. Org. 1970, 4, 588–592, doi:10.1039/J29700000588.

Yu, Z.X.; Caramella, P.; Houk, K.N. Dimerizations of Nitrile Oxides to Furoxans Are Stepwise via Dinitrosoalkene Diradicals: A Density Functional Theory Study. J. Am. Chem. Soc. 2003, 125, 15420–15425, doi:10.1021/ja037325a.

We believe these references adequately address the historical and mechanistic understanding of nitrile oxide dimerization and provide a solid foundation for the discussion in our manuscript.

Compounds metrics (including data from the NMR, IR and HRMS analysis as well as m.p's and informations about the yields) should be collected in the main file of the manuscript.

We have consolidated all compound metrics, including NMR, IR, melting points (m.p.), and information about the yields, into the main file of the manuscript. However, we did not include HRMS data as we do not have access to a spectrometer. Obtaining these spectra would require collaboration with another university, which could take up to six months.

Lastly, some technical errors (sentences such as "Error! Reference source not found.") should be corrected along to all parte of the manuscript.

I have thoroughly reviewed the manuscript and did not find any instances of "Error! Reference source not found."

Sincerely,

Assoc. prof. Mária Vilková

Reviewer 3 Report

Comments and Suggestions for Authors

Vilkova and coworkers reported their findings of 1,3-dipolar cycloaddition of nitrile oxides. The author take a lot of time to analyze the NMR spectrum of the obtained compounds to determine their structures, This analysis process is impressive. However, this work is not novelty, the racemic cycloaddition is a common reaction, and the main work is about the cycloaddition of a special alkene and nitrile oxides. It doesn’t have any significant contribution to this area. Also this work is not well presented, what’s the structure of 4? There should be a scheme in the introduction part. In summary, this work can’t be acceted.

Author Response

Dear Reviewer 3,

I hope this letter finds you in good health and spirits. I am writing to provide an update on our manuscript entitled “Spectral Assignment in the [3+2] Cycloadditions of Methyl (2E)-3-(acridin-4-yl)-prop-2-enoate and 4-[(E)-2-Phenylethenyl]acridin with Unstable Nitrile N-Oxides” to the special issue New Insights into Nuclear Magnetic Resonance (NMR) Spectroscopy for consideration for publication in Molecules.

Thank you for your detailed feedback on our manuscript. We appreciate your recognition of the thorough NMR analysis we performed to determine the structures of the obtained compounds. Regarding the novelty of our work, we understand your concern about the common nature of racemic cycloaddition reactions. However, our study specifically focuses on the cycloaddition of a unique set of alkenes and nitrile oxides, which we believe provides valuable insights into their reactivity and potential applications. We respectfully disagree with the assessment that our work lacks significant contribution to the field, as it expands the understanding of these specific reactions.

In response to your comments on presentation, we acknowledge the need for improvement. We will include a scheme in the introduction to clearly present the structure of compound 4 and provide a better overview of our study.

We appreciate your constructive criticism and will work on enhancing the clarity and presentation of our manuscript.

Sincerely,

Assoc. prof. Mária Vilková

Round 2

Reviewer 3 Report

Comments and Suggestions for Authors

The revised paper looks much better now. Scheme 1 makes the paper well understood. The major issue is still the novelty. In the reported cycloaddition, a special alkene was used. What’s the special about it? Special alkene would not make this reaction general. Besides, the reaction needs 8 days to finish, what if some catalysts were added, moreover, the reaction is racemic. I don’t think this transformation is significant from a synthetic view. However, the NMR analysis of the product is really impressive. In summary, this paper can’t be accepted.

Author Response

Dear Reviewer,

Thank you for your feedback on our revised manuscript. We appreciate the time and effort you put into reviewing our manuscript.

We are glad to hear that you find Scheme 1 useful for understanding the article and that you are impressed with the NMR analysis of the product. We would like to address the main concerns you have expressed regarding the novelty and general applicability of the reported cycloaddition reaction.

Firstly, we acknowledge that the particular alkene used in our study may limit the generality of the reaction. However, the unique reactivity of this alkene offers valuable insight into the reaction mechanism, which we believe contributes to the fundamental understanding of cycloaddition chemistry. Our goal at this stage was to obtain product in sufficient amounts rather than to propose a widely applicable synthetic method.

Regarding the reaction time, we agree that eight days is quite lengthy. We are currently investigating the potential benefits of various catalysts to speed up the reaction. This ongoing research will be the focus of future publications where we hope to report significantly improved reaction conditions.

We also acknowledge the limitation of the racemic reaction. Our main goal in this study was to achieve a successful transformation and thoroughly analyze the product. Enantioselective versions of the reaction are certainly of interest, and we plan to explore chiral catalysts and other methods to address this in our future research.

The isolation of the products presented significant problems that prevented us from submitting samples for biological evaluation. Despite these difficulties, we believe that our findings regarding the reaction mechanism and our detailed NMR analysis are valuable contributions to the field.

It is important to emphasize that in today's research environment, structural analysis of products is often not as thorough as it should be. There are numerous publications with incomplete or incorrect structural characterizations. Just recently, I came across an article in the journal Results in Chemistry, where the structures were not determined correctly despite the 13C NMR spectra providing clear evidence of a completely different structure of the products. The authors ignored resonance lines in the spectrum that did not fit their proposed structures, and this went unnoticed by reviewers. By providing an in-depth NMR analysis, we aim to highlight the importance of accurate structural elucidation, which we believe is crucial for progress in synthetic chemistry.

We believe that the content of our manuscript is suitable for this special issue, given its focus on rigorous NMR structural analysis. We hope these points address your concerns and demonstrate the value of our work.

Thank you once again for your constructive feedback.